# TimesVector-Web: A Web Service for Analysing Time Course Transcriptome Data with Multiple Conditions

**DOI:** 10.3390/genes13010073

**Published:** 2021-12-28

**Authors:** Jaeyeon Jang, Inseung Hwang, Inuk Jung

**Affiliations:** Department of Computer Science and Engineering, Kyungpook National University, Buk-gu, Deagu 41566, Korea; wodus1035@knu.ac.kr (J.J.); his0405@knu.ac.kr (I.H.)

**Keywords:** web service, time course, clustering, gene expression pattern

## Abstract

From time course gene expression data, we may identify genes that modulate in a certain pattern across time. Such patterns are advantageous to investigate the transcriptomic response to a certain condition. Especially, it is of interest to compare two or more conditions to detect gene expression patterns that significantly differ between them. Time course analysis can become difficult using traditional differentially expressed gene (DEG) analysis methods since they are based on pair-wise sample comparison instead of a series of time points. Most importantly, the related tools are mostly available as local Software, requiring technical expertise. Here, we present TimesVector-web, which is an easy to use web service for analysing time course gene expression data with multiple conditions. The web-service was developed to (1) alleviate the burden for analyzing multi-class time course data and (2) provide downstream analysis on the results for biological interpretation including TF, miRNA target, gene ontology and pathway analysis. TimesVector-web was validated using three case studies that use both microarray and RNA-seq time course data and showed that the results captured important biological findings from the original studies.

## 1. Introduction

Times-course analysis of gene expression data can be advantageous for revealing modulating gene expression patterns of certain biological mechanism across time. It is a common practice to search for significantly differentially expressed genes (DEGs) between two conditions, for example, a cohort of normal versus cancer patients. Such analysis is tailored to observe the transcriptomic difference within a single snapshot of the current gene expression status in a pairwise manner. Thus, the dynamics or ongoing transition of important gene regulatory functions may be overlooked.

Time course analysis data is generated to observe any significant modulation of gene expression or other omics data to explain the biological phenomena of interest in respect to the condition of the experiment. Time course analysis can become a complex procedure and can produce different results depending on the researchers perspective of the analysis. For example, when adopting the traditional DEG analysis, we are opted to compare a combination of time point pairs. Such approach requires post analysis of the result, since the results of each DEG pair itself is not sufficient for interpretation. They need to be integrated afterwards to derive biologically meaningful results. In terms of experimental setting, if time course data is generated for a single condition, identified gene expression patterns are expected to be able to explain the response to the related condition [1,2,3]. In the other hand, if time course data is generated for multiple conditions, finding gene expression patterns that significantly differ (or similar) between the conditions are the main interest of the analysis [4,5,6].

Collectively, time course data allows us to search for genes that yield significantly different expression patterns across time. Here, the gene expression patterns can be used to comprehend the biological mechanisms with greater detail. For example, we can observe how genes of a certain pathway respond to some external stress across time. More importantly, we can pinpoint the time when such genes started to respond to the stress. However, time course data is expensive and choosing the time points for sampling is a non-trivial task, requiring expert knowledge and careful pre-experimenting on putative or known condition responsive genes. Without such careful design, important gene expression modulation may not be captured within the selected time course. Until now, many time course analysis methods for identifying differentially expressed genes were developed. However, due to the difficulty of the interpretation of the result or usage of the tool, many studies base their analysis on DEG methods [7]. For biologists, offline software may be difficult to use, which may sometimes not be compatible with certain systems or even become outdated. Nevertheless, only a few tools provide a web-based time course gene expression analysis platform. GEsture [8] is a web-based tool that helps to search for user specified gene expression patterns from time course gene expression data. Hence, instead of performing clustering in a traditional way, the results depends on the user specified search pattern. Thus, patterns different from the specified input are possible missed.

To alleviate the burden for performing the rather difficult time course analysis, we developed an easy to use web service for analysing time course gene expression data with multiple conditions. The web service implements the algorithm of TimesVector [4], previously developed by us, with several enhancements in means to simplify the analysis procedure and providing downstream analysis for various biological interpretation. The TimesVector-web is freely available at https://cobi.knu.ac.kr/webserv/TimesVector-web (accessed on 26 November 2021). TimesVector is a time course gene expression analysis tool that aims to search for gene clusters that exhibit differential expression patterns across multiple conditions in time. Since there may be multiple conditions and multiple time points, the data forms a three dimensional structure (i.e., gene, time and condition). Since sub-space clustering (or bi-clustering) on two dimensional is NP-hard [9,10], three dimensional clustering is a difficult task. So, the samples are concatenated across time to convert the 3D data into 2D data. On the transformed 2D data, the spherical clustering is performed to identify gene clusters that have high similarity within the same cluster and low similarity across the other clusters. Afterwards, the samples are split per condition and tested for pattern difference using mutual information. The study in [4] may be referred for further details.

The functions of the web-based TimesVector are as below:A simple user interface for uploading data and parameter configurationGene filtering functionSupport of microarray and NGS dataGene expression normalizationSupport of sample replicatesA quick test for choosing a sub-optimal *K* (i.e., the number of clusters or patterns)Interactive result page for visualizing and selecting the searched gene cluster patternsBiological downstream analysis including GO (Gene Ontology) analysis, pathway analysis, identification of Transcription Factors (TF) and putative miRNA targets

The web-service mainly focuses on providing (1) intuitive and easy click-based analysis interface, (2) visualizing the clustering results and (3) continuous downstream analysis further biological comprehension of the cluster results. Based on the pattern, each cluster is categorized as one of the following: Differentially Expressed Pattern (DEP), at least One Differentially Expressed Pattern (ODEP) and Similarly Expressed Pattern (SEP) clusters. Here, a DEP cluster is defined as a cluster where the genes have similar expression patterns within each condition but have different expression patterns across all the conditions. ODEP is a cluster where the pattern is different in only one condition. At last, the SEP cluster is comprised of genes whose expression patterns is similar across all the conditions. To interpret the searched DEP, ODEP or SEP clusters, candidate TFs and miRNA target genes are identified using external TF and miRNA databases. This is because TFs and miRNAs are well known regulators influencing the gene expression. Furthermore, gene set enrichment analysis is performed on the DEP, ODEP and SEP type clusters for biological interpretation.

As a guide to use the TimesVector-web, we present the analysis results using three case studies that use both microarray and RNA-seq time course data.

## 2. Materials and Methods

The TimesVector-web is executed by a number of procedures in a sequential manner. It starts from preparing the input data files and finishes with the output results. Each procedure is described in detail in the latter section followed by two tutorials in the website. The workflow of TimesVector-web is shown in Figure 1.

### 2.1. The User Interface

The TimesVector-web is a web service that allows users to analyse microarray and high-throughput time course gene expression data in means to detect differential and similar gene expression patterns across multiple conditions. Thus, the main objective is to detect the differentially modulating gene expression patterns between two or more experimental conditions (e.g., conditions, stress or phenotypes). Its simple user interface and the downstream analysis function for biological interpretation of the cluster results makes it easy to perform time course analysis without the need of expert computing knowledge. Furthermore, TimesVector-web requires only a few parameters (i.e., number of time points, conditions, clusters and replicates) as input. For sub-optimal results, the number of clusters can be automatically estimated by the *K*-test provided by the web-service.

The user interface mainly consists of two panels; the input panel and the result panel. In the input panel, the time course gene expression data can be uploaded followed by options for setting the parameters required for preprocessing and clustering the input data as shown in Figure 2A. First, to perform the analysis, the time course gene expression data need to be uploaded. Since we are interested in comparing time-course data of multiple conditions, the user may select two or more time-course data files, each corresponding to a specific condition. For example, the data from GSE4324, used for our case study 2, is comprised of four conditions, which are two male and two female mice with state of gonadectomy. A time-course data of four time points is present for each of the four conditions. Thus, in this case, the number of conditions is 4 and number of time points is 4. Furthermore, this data sets have three replicates in each time point. Hence, the number of replicates is set to three. These parameters are automatically detected by the web-service by parsing the header information of the given input files.

The process of preparing the input file is described in detail in the latter section “Preparing the input file” (Section 2.2). Second, users need to set parameters related to data preprocessing (i.e., normalization, data type indication and gene selection) and clustering. As aforementioned, the number of time points, conditions and replicates are automatically detected by the uploaded time-course data files. As any traditional *K*-means based clustering algorithm, the number of clusters to be found need to be provided as the input parameter *K*. Parameter *K* is an integer, that indicates the number of time-course patterns to be identified. However, since there may be patterns that are not significantly different across the conditions, only clusters with a significant *p*-value are reported. Collectively, the number of reported clusters are usually smaller the *K*. Thus, it is suggested to start the analysis with a large *K* (e.g., 100 or higher). *K* is the most important parameter, which has the greatest impact on the result. The user may run the analysis with different *K*’s. In case it is difficult to find a good *K*, we provide a *K*-test function in means to help users to select a good *K*, which will be discussed in section “Finding an optimal *K* for clustering” (Section 2.3) in more detail. The user can select the organism of interest that fits the data in order to obtain biologically meaningful clusters. Finally, the user can execute the analysis by clicking the Run button.

Once the executed analysis completes, the user will be directed to the result panel as shown in Figure 2B. The result panel consists of three sub-panels; (1) summary statistics of the data and result, (2) identified cluster patterns and (3) lists of gene regulatory miRNAs and TFs followed by tabs showing the gene set enrichment analysis results of the identified cluster patterns. The details of the result page will be discussed with more detail in section “Functional analysis of the clusters” (Section 2.5). For each analysis, a unique shortcut code is provided to the user upon execution. The runtime of the analysis varies from several minutes up to 30 min which depends on the value of *K*. The majority of time is spent on plotting the cluster patterns. For example, the analysis of case study 1, which is discussed in the result section, took 12 min to complete, while case study required up to 30 min. The runtime is expected to shorten significantly with smaller number of genes, hence filtering non-protein coding genes is advantageous for performing a brief analysis. Thus, to prevent the user from waiting, the shortcut key can be used to track the progress or revisit the results by entering the shortcut code into the shortcut panel located under the main input panel as shown in Figure 3.

### 2.2. Preparing the Input File

The time course expression file needs to be prepared in a pre-defined format before uploading it. TimesVector-web accepts both RNA-seq and microarray type gene expression data, which must be stated by setting the “Data type” option in the input panel. The units of gene expression is not limited to RPKM, FPKM, TPM or raw read count. TimesVector-web performs clustering on unit vectors, thus normalizes the data during the preprocessing stage, which includes sample and quantile normalization. While we recommended to provide raw read count data, the performance of the results when using different RNA-seq gene expression units was marginal. For microarray type data, the gene expression matrix is expected to be already normalized. However, the data may be normalized by the user using their own methods, in which case the normalization option shall be deselected in the input panel. TimesVector-web supports samples with replicates. Simply, the average expression value of genes of the replicates are computed. Finally, the header of the input file should follow a specific syntax where all columns are separated by tabs. The first element in the header must be named GeneID and the names of the remaining columns should have a format as follows: ‘Condition’_‘TimePoint’_‘replicate’ (e.g., DV10_Day2_rep1). Here, Condition refers to the name of the condition or phenotype of the sample, TimePoint refers to the time point of the sample and replicate indicates the name of the replicate of the sample. The number of replicates is expected to be the same in all conditions and time points. If no replicates are present, the last replicate field shall be omitted.

An example input file is shown in Table 1, which has four conditions (e.g., intactmale, gdxmale, intactfemale, gdxfemale), four time points (e.g., day0, day3, day7, day14) and three replicates (e.g., rep1, rep2, rep3) per time point.

Optionally, the user may constrain the gene list to use only protein coding genes in case non-coding genes are not of interest. If the “Use only protein coding genes” option in the input panel is selected, only genes that are labeled as protein coding are selected from the input file. The protein coding label annotation data from BioMart [11] is used.

Regarding the gene IDs in the input file, TimesVector-web accepts Ensembl ID, Refseq ID and gene symbols for gene names. The gene IDs are converted and matched internally using the gene annotation data from the Ensembl Biomart [12].

### 2.3. Finding an Optimal K for Clustering

To analyse the gene expression pattern in data, it is important to select an appropriate number of *K* clusters. The appropriate number of clusters here is when the number of clusters is the smallest, the distance of genes within the same cluster is small and the distance of genes across different clusters is large.

We used the traditional elbow method [13] to help users choose the appropriate number of clusters in case it is difficult to choose one. In this method, the selection criteria for determining a good *K* is based on selecting *K* from a point that minimizes the number of clusters while minimizing the inertia. However, when using the basic elbow method, the graph’s appearance is inconsistent due to the fundamental problem of *K*-means clustering where the initial centroids are randomly positioned. To avoid such problem, *K*-means clustering was run several times and used the average value of total within-clusters sum of squares.

TimesVector-web provides $K_{max}$ parameter as an option when performing *K* test. This parameter is for setting the range of *K* that the user wants to select since the result of *K* test can vary depending on the size or characteristics of the input data. The users can indicate the range of *K* to be searched by setting the “Max K” parameter in the input panel. The *K* test will test a range of *K*s starting from 2 to $K_{max}$ in steps of $K_{max}$/10. Figure 4 shows the result of the *K* test after setting the $K_{max}$ parameter to 800, where the optimal *K* was 170. Here, the *x*-axis is the varying *K* and *y*-axis is the sum of square distance of genes to their centroid (i.e., sum of intra-cluster distance of all clusters). A *K* of 170 does not mean that the result will output 170 clusters. Instead, from the 170 clusters, each cluster will be tested for statistical significance in pattern difference across the conditions. Thus, a much smaller number of clusters may be the result of the analysis even with such a high *K*. However, when choosing a rather too large *K*, one may get many small clusters that actually have similar patterns, thus redundant. To avoid such redundancy, it is recommended to start with a small *K* and increase if the biological results are not satisfactory.

### 2.4. Detecting Differential Gene Expression Patterns

For detecting gene clusters with significant differential expression patterns (DEP), TimesVector [4] is used, which clustering algorithm is based on the spherical *K*-means algorithm [14]. The result of *K*-means type clustering methods are affected by the initially set *K* centroids since in many cases they are set in random manner. To improve the performance, often the K-means++ methods [15] is applied to approximately find good initial cluster centroids or seeds. The K-means++ searches for centroids that are most apart from each other by computing the distance between the data points. We reduced the execution rate by searching for only 10% of the total data points which shows good cluster results requiring very short time even with a large *K* (e.g., 78 s for obtaining the results in Figure 4).

Each cluster is classified as either a DEP, ODEP or SEP. A DEP cluster is a set of genes that have similar gene expressions within conditions, but are different in other all conditions. An ODEP cluster is a set of genes where gene expression pattern in at least one condition is different from others. An SEP cluster is a set of genes that share a common expression pattern across all conditions. Only clusters with patterns of statistical significance are selected as the final result. The clustering result showing DEP, ODEP and SEP clusters is shown in Figure 2B above.

### 2.5. Functional Analysis of the Clusters

The genes in each detected cluster serve as sources for biological interpretation of the conditions or phenotypes of interest. Our web service provides a five of analysis panels for visualizing gene cluster patterns, performing gene set enrichment analysis and searching for gene regulatory elements, such as TFs and miRNAs. The functional analysis tabs are located at the bottom of Figure 2B, where one may choose to view (1) the expression pattern of the selected cluster, (2) the gene list of the cluster, (3) the list of TFs present in each cluster, (4) list of genes and their target miRNAs and (5) gene set enrichment analysis for each cluster type (i.e., DEP, ODEP and SEP).

#### 2.5.1. Cluster Pattern

Starting from the left, the first tab “Cluster pattern” shows the detailed expression pattern of the selected cluster. The left image represents the average expression pattern of each condition and the right depicts the expression pattern of each gene of the cluster (Figure 5). Here, we can observe that the genes are significantly expressed in day 7 specific to the condition color coded in yellow.

#### 2.5.2. Gene List

The second tab, “Gene list”, lists genes belonging to the selected cluster, which can be selected in the “Cluster” panel within the result page. An example gene list of a cluster is shown in Figure 6A. Here, the annotation of gene type and function are provided. If the input file contains gene names that are not available in the Ensembl DB, the analysis functions may result in an empty list.

#### 2.5.3. TF Analysis

The third tab, “TF”, lists the TF genes present in the selected cluster as shown in Figure 6B. TF genes are searched in the Cis-BP database, which is a library of TF DNA binding motifs and their specificity [16]. Since gene expression is promoted by TFs, the TFs in a cluster may be candidate gene regulatory factors, which may be responsible for the expression pattern of the cluster.

#### 2.5.4. miRNA Analysis

Similar to TFs, putative gene targeting miRNAs are lists in the “miRNA” tab (Figure 6C,D), which lists the genes that are known to be targeted by at least one miRNA with high confidence. The miRNA and gene target pair information curated in miRDB [17] was used. Currently, mirDB supports only five types of genomes vertebrates human, mouse, rat, dog and chicken, thus this tab will not output results for other genomes.

#### 2.5.5. Functional Enrichment Analysis

At last, the “g:Profiler” tab provides the result of gene set enrichment based analysis to detect significant GO terms, pathways, TF binding sites and miRNA targets (Figure 7) using g:GOST from the g:Profiler [18] tool. Here, the *x*-axis of each barplot represent the $-log_{10}\left(p.adj\right)$ of the enrichment analysis results. Normally, a cluster is comprised of not many genes, which will result in empty results when performing gene set enrichment analysis. Hence, genes were aggregated for each cluster type (i.e., DEP, ODEP or SEP), which are given as input to the enrichment analysis.

## 3. Results

We used TimesVector-web to analyze multi-class time course data of three case studies and show that TimesVector-web was able to capture biologically meaningful results that agree with the outcomes reported in the original studies. The dataset of the three case studies includes both microarray and RNA-seq data and have different characteristics, such as time points, conditions and number of replicates which are shown in Table 2. These datasets were collected from the GEO [19] database. The required input parameters for each case study can be loaded by clicking on the case study example buttons in the “Case study examples“ panel in the website. Associated time-course data files will be downloaded as well, which are required to be uploaded as data files.

### 3.1. Case Study 1: Time Course Analysis of Five Yeast Strains during Alcoholic Fermentation (GSE11651)

The dataset of this case study was collected from the research in [20]. Here, the gene expression of five different yeast strains (i.e., Saccharomyces cerevisiae) was profiled by microarray, which were taken at three different time points for each strain during alcoholic fermentation. Fermentation was performed independently of three barrels and three repeated experiments were performed per barrel. Namely, this data set consists of three replicates, five conditions (i.e., strains), and three time points. A total of 10,639 genes were present in the dataset, which were subject to analysis. Non-protein coding genes were kept and normalization was not performed since the provided data was already preprocessed. Since three replicates were present per sample, the number of replicates was set to 3. The *K* was set to 30 according to the *K* test result.

TimesVector-web found a cluster, which expression pattern was the highest specific to the DV10 strain as shown Figure 8). This cluster had the largest number of genes compared to the other clusters. Furthermore, the “Aromatic compound biosynthesis process” GO term had a significant *p*-value of 0.0125 which was also reported in the original study. Collectively, the clusters were able to partially reproduce the results of the study.

### 3.2. Case Study 2: Analysis of Malaria Infection Responsive Genes in Mice with Different Gonadectomy and Sex Conditions (GSE4324)

Here, we collected time course gene expression data from GSE4324 [21]. Here, the study investigated the differentially expressed genes in respect to gonadectomy status (i.e., intact and sex in time course manner). Thus, the authors searched for gene expression pattern signatures that was caused by gonadectomy (gdx) in both male and female mice. Hence a total of four conditions were present for which the gene expression were measured at 0, 3, 7 and 14 days post malaria infection. Three replicates were present for each condition time point and a total of 18,116 genes were subject for analysis. Similarly, the *K* recommended by the *K* test, which was 170. However, the functional results of the detected clusters were not sufficient for interpretation and also agreed with the original study. The *K* was increased to 500 to detect more cluster patterns in means to improve the functional result. As a result, TimesVector-web detected 2 DEP, 9 ODEP and 21 SEP gene clusters, which contained 31, 1543 and 3258 genes respectively, which well captured related biological signals inline with the previous study.

In the most ODEP clusters, genes of gonadectomized samples and intact female sample have similar expression patterns, but genes of intact male sample have different patterns from them. Furthermore, in these clusters, the difference in gene expression level was the largest at 7 days among several time points. The related study showed that the levels of parasitemia in gonadectomized male sample are marginally higher than those of intact males, intact females, and gonadectomized females at 7 days. In the DEP and ODEP clusters, we found that the gene expression of gonadectomized female sample was higher than that of other samples.

In addition, we obtained biologically interpretable results from certain cluster patterns through the functional enrichment analysis provided by the TimseVector-web. Other study that conducted GO (Gene Ontology) analysis with the same data showed GO terms related to immune response in clusters that are upregulated and downregulated in intact male sample [22]. Also, we found that kind of conclusion too. Several GO terms related to immune response are founded in the GO biological processes of the DEP such as ‘defense response (GO:0006952)’, ‘response to stress (GO:0006957)’ and ‘immune response (GO:0006955)’. Detailed GO biological processors related to the DEP can be found in Figure 9.

### 3.3. Case Study 3: Results for Rice Data with Different Root Structures

The third case study is the rice root RNA-seq data GSE92835. Data is the rice root to know the process by which formation of nodule-like structures(NLS) are induced by plant hormones with three time points 0, 7 and 14 days. Especially conditions are inducing or not NLS by applying synthetic auxin, 2,4-dichlorophenoxyacetic acid (2,4-D) to root of Oryza sativa. Also, each samples in this data set included three biological replicates. Namely, two conditions, and three different time points were present. As a result of performing *K*-text on the corresponding data set, the appropriate number of clusters was recommended as 30.

In the reviewed papers, biological characteristics were indicated by comparison for each time point pairwise [23]. By comparing data on days D1, D7, and D14, respectively, differentially expressed genes (DEGs) and upregulated and downregulated genes were subject to enrichment analysis using singular enrichment analysis (SEA) of agriGO [24]. As a result of the functional enrichment analysis, the “Secondary metabolic progress” biological process had a significant *p*-value by comparing all the time points of DEP (Figure 10). Also GO terms related to the “Cell wall” of the cellular component, were identified in DEP clusters. In ODEP clusters, the “Metabolic process”, “Biological regulation”, and “Regulation of cellular process” were identified, which were also reported in the original study. All of the TFs identified in the related studies AP2/ERFs, auxin response factors (ARFs), lateral organ boundaries (LOB), NAC domain-containing proteins, WUSCHEL-related homeobox proteins, and GRAS family were also identified by TimesVector-web. In addition, important TF domains that are related to root structuring, such as B3, BLHL, R2R3, MYB and WRKY were further detected.

## 4. Conclusions

Here, we present the TimesVector-web tool that provides an easy to use interface for the analysis of time course gene expression data across multiple conditions. The objective is to detect gene express patterns that are significantly different or similar across the conditions. Also, for biological interpretation, several downstream analysis functions are provided, such as TF, miRNA target analysis and gene set enrichment analysis for detecting significantly enriched GO terms and pathways. The tool is a web based service which we hope to alleviate any difficulty required to perform time course analysis manually.

Using three case studies, we showed that TimesVector-web was able to handle various types of data (i.e., microarray, NGS), and datasets with different number of time points, conditions and genes. More importantly, TimesVector-web was able to reproduce important results reported in the original studies of the three case studies.

The TimesVector-web is to be extended to further handle single-cell RNA-seq time course data.

## Figures and Tables

**Figure 1 genes-13-00073-f001:**
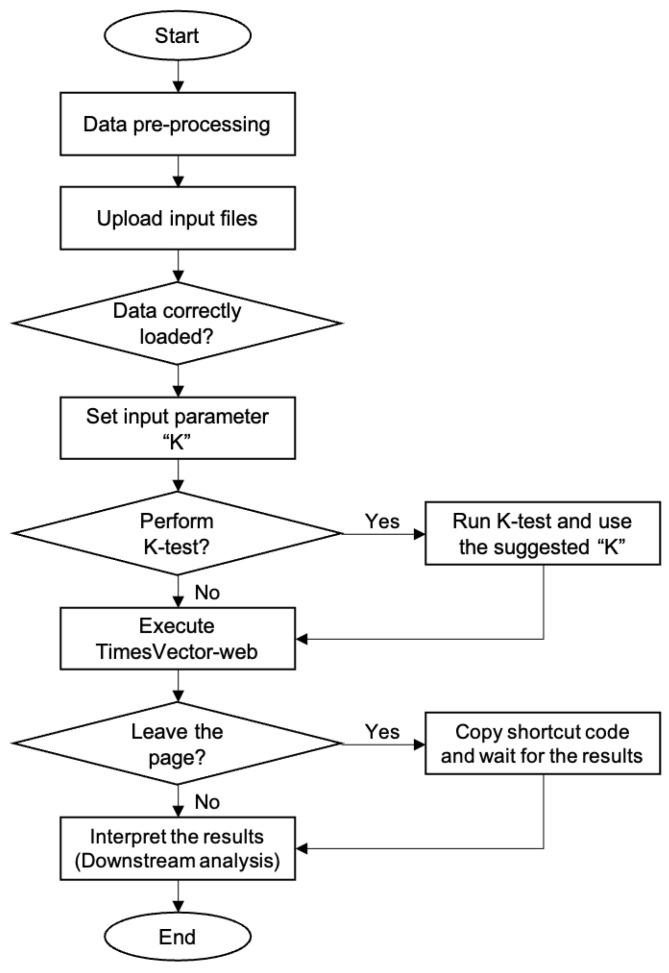
The workflow of TimesVector-web.

**Figure 2 genes-13-00073-f002:**
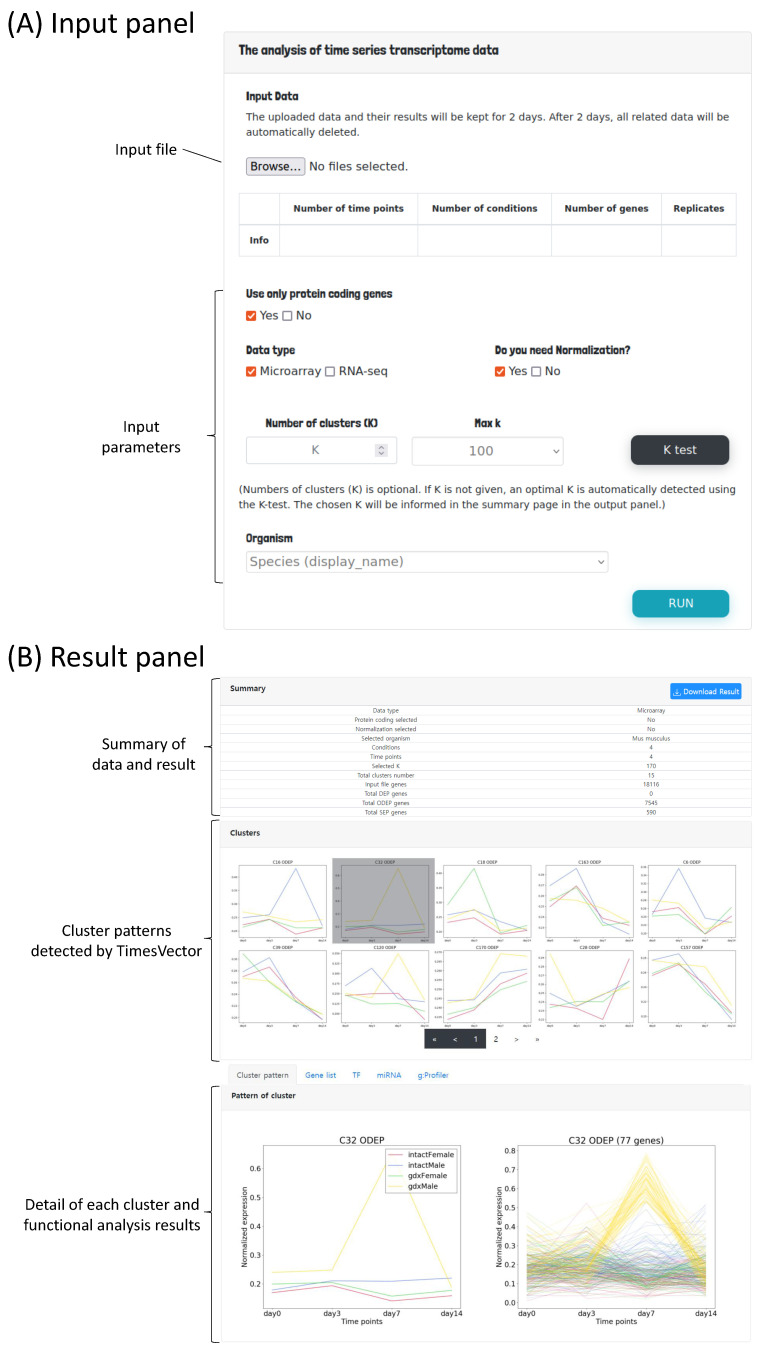
The user interface of TimesVector-web. (**A**) The user input panel for uploading the time course gene expression data and setting input parameters. (**B**) The results panel for viewing the clustering results and performing downstream biological functional analysis.

**Figure 3 genes-13-00073-f003:**
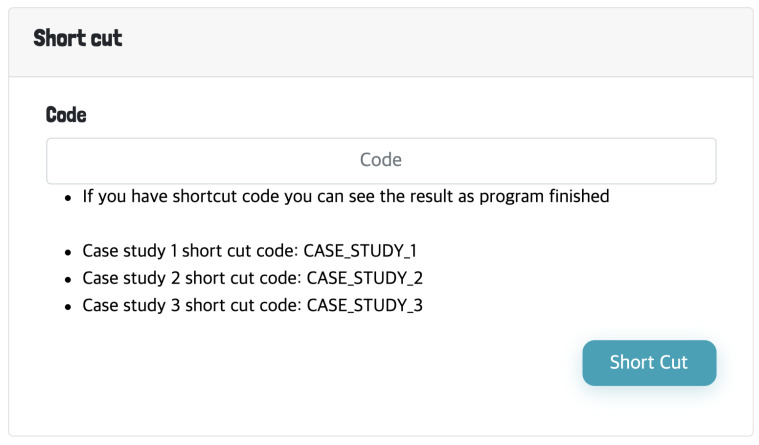
The shortcut panel for searching previous analysis results.

**Figure 4 genes-13-00073-f004:**
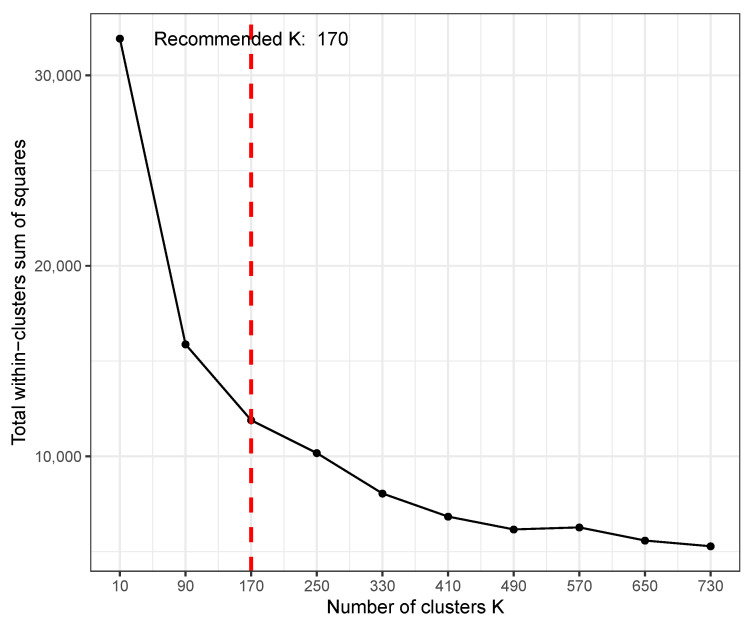
An example result of *K*-test using time course gene expression data with five conditions and three points.

**Figure 5 genes-13-00073-f005:**
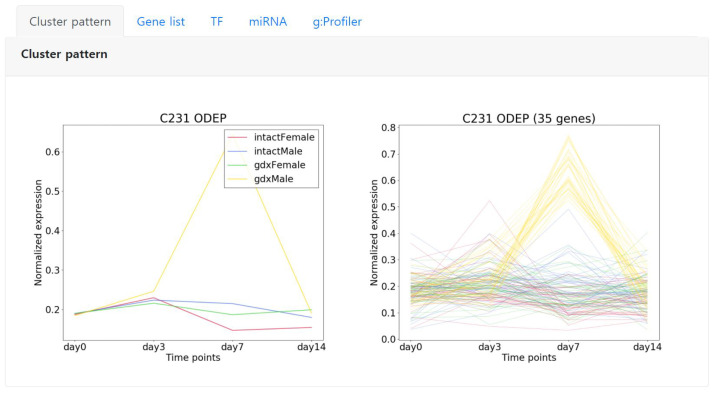
An example showing the expression pattern of an ODEP type cluster C231 that is comprised of 35 genes. Left shows the average pattern of each condition, right shows the expression pattern of each gene in the cluster. The conditions are color coded.

**Figure 6 genes-13-00073-f006:**
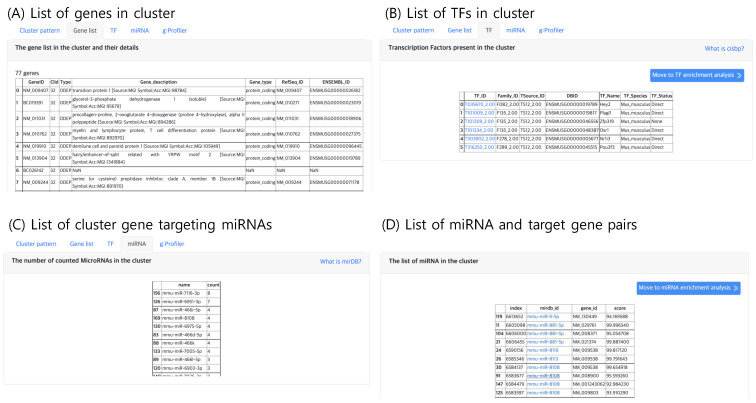
The result tabs showing (**A**) the gene list in a cluster (**B**) TFs present in a cluster (**C**) Number of counted miRNAs in a cluster and (**D**) the list of miRNAs and their target genes in a cluster.

**Figure 7 genes-13-00073-f007:**
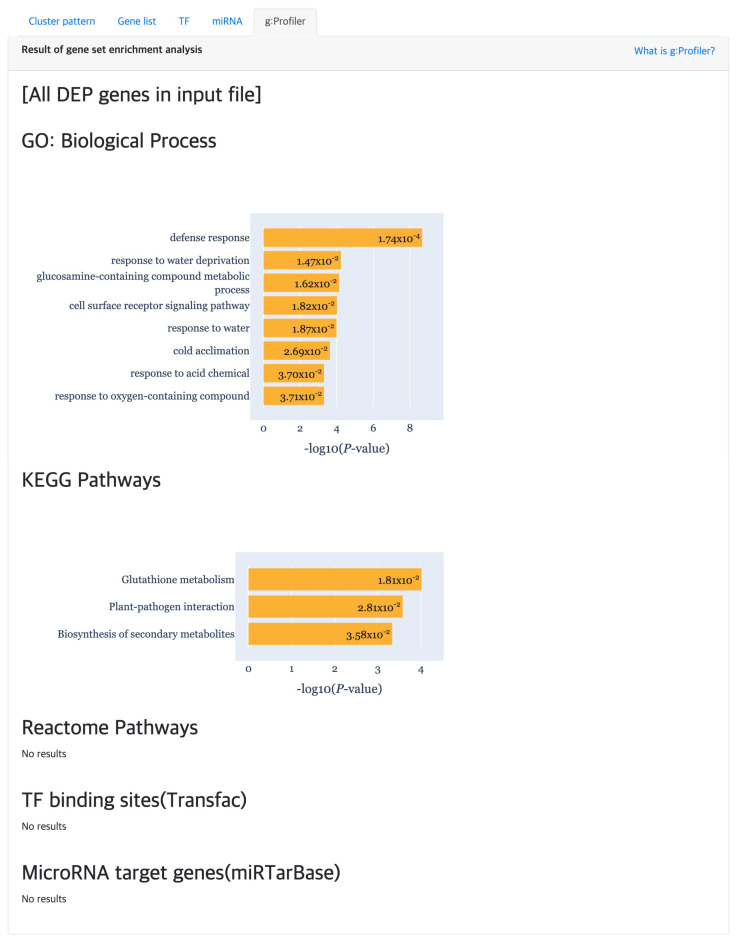
The gene set enrichment analysis results for detecting significant GO terms, pathways, TF binding sites and miRNA target using g:Profiler.

**Figure 8 genes-13-00073-f008:**
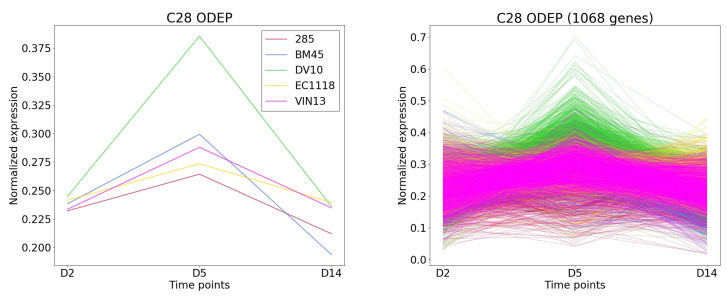
The cluster and gene patterns of cluster 28 which express gene highest in DV10 strain for case study 1.

**Figure 9 genes-13-00073-f009:**
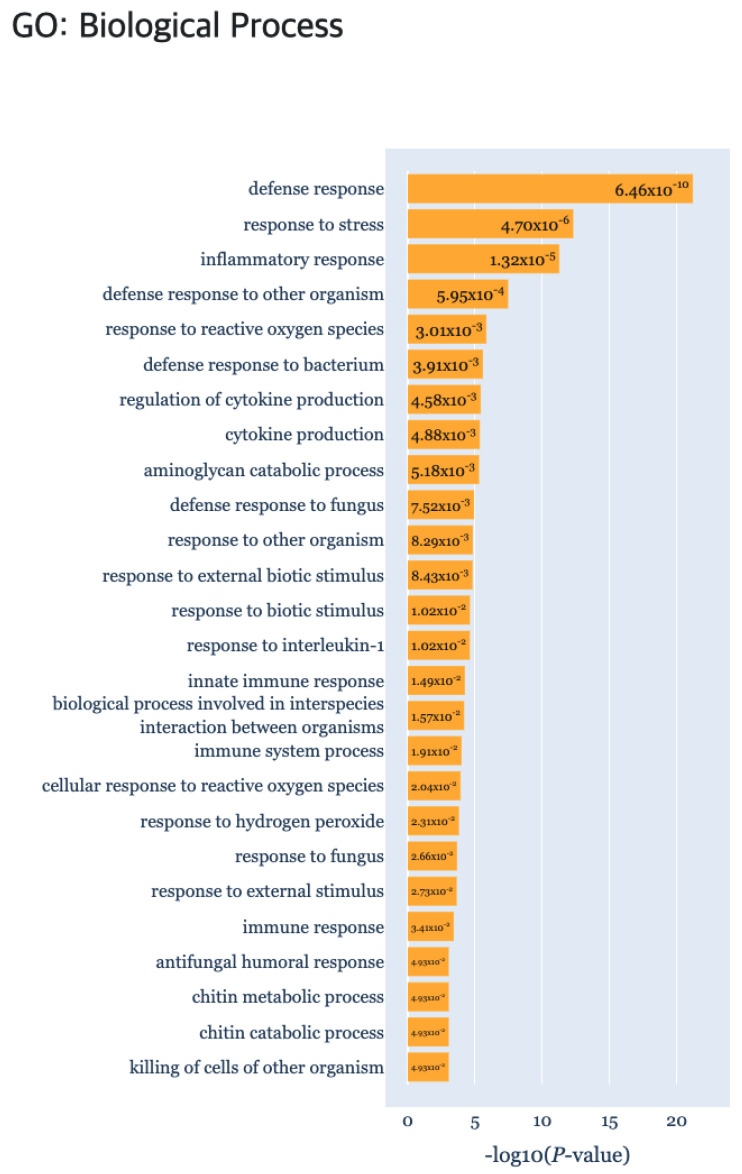
The gene set enrichment analysis results for Case study2 detecting significant GO: Biological Process using g:Profiler.

**Figure 10 genes-13-00073-f010:**
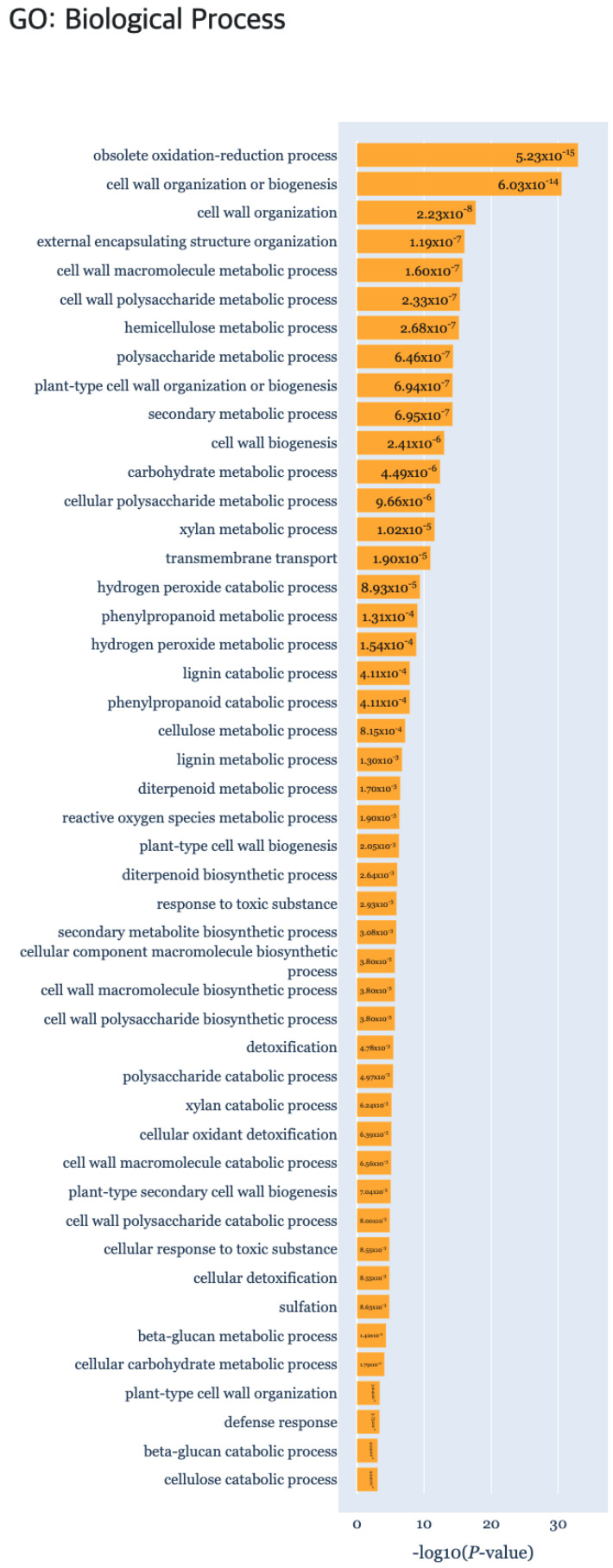
The gene set enrichment analysis results for Case study3 detecting significant GO: Biological Process using g:Profiler.

**Table 1 genes-13-00073-t001:** An example time course gene expression input file of single condition.

GeneID	intactmale_day0_rep1	intactmale_day0_rep2	intactmale_day0_rep3	intactmale_day3_rep1	…	intactmale_day14_rep1	intactmale_day14_rep2	intactmale_day14_rep3
NM_013477	1281.1	1158.5	1455.8	1522.7	…	1822.5	1205.3	1601.6
NM_020585	1325.8	1569.2	1258.0	1250.5	…	1804.1	1139.4	1393.6
NM_133900	434.8	450.7	371.9	492.8	…	297.5	384.8	402.4
NM_021789	967.1	954.1	1047.2	1222.4	…	767.9	1042.7	968.8
...	…	…	…	…	…	…	…	…
NM_007422	1542.2	1525.0	1712.7	1689.3	…	1625.7	1611.5	1529.3

**Table 2 genes-13-00073-t002:** Dataset statistics of the three case studies.

Data	# of Genes	Time Points	Conditions	Organism	Data Type
GSE11651	10,639	3	5	Yeast	Microarray
GSE4324	18,116	4	4	Mouse	Microarray
GSE92835	66,860	3	2	Oryza sativa	RNA-seq

## Data Availability

The TimesVector-web service is freely available at http://cobi.knu.ac.kr/webserv/TimesVector-web (accessed on 26 November 2021).

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
