# Peer review of "TimesVector-Web: A Web Service for Analysing Time Course Transcriptome Data with Multiple Conditions"

_genes, 2021, doi:10.3390/genes13010073_

Round 1

Reviewer 1 Report

The web service TimesVector-web implements the algorithm of TimesVector (published on the journal of Bioinformatics) with several enhancements in means to simplify the analysis procedure and providing downstream analysis for various biological interpretation. This paper provided three case studies that use both microarray and RNA-seq time course data and showed that the results captured important biological findings from the original studies.

I have some minor suggestions.

  • Authors should provide more examples on the website.
  • Provide one or more example buttons to load examples and parameters with one click.
  • Button text is sometimes related to the operating system. However, it is better to change the text of the button in the screenshot of the paper (input file button) to English rather than Korean.

Author Response

Dear Editor and Reviewers,

We are very grateful for having the chance for revising the manuscript. The comments of the reviewers were very constructive which helped us improve the quality of the study. Submitted is the revised manuscript entitled "TimesVector-web: A web service for analyzing time course transcriptome data with multiple conditions" by Jaeyeon Jang, Inseung Hwang and Inuk Jung.

Due to the limitation of text, the response to the reviewers were written in attached PDF file. We made our best to to address all the comments from the reviewers. The revised manuscript incorporates the responses and amendments to all the comments from the reviewers.

We hope that the editor and reviewers find our amendments satisfying.

Best regards,

19th Dec. 2019
Inuk Jung

Reviewer 2 Report

The manuscript presents a web tool for analyzing time-course and multi-condition trascriptome data. The user interface of the web tool is clean and clear, however, the instruction and description of the tool is very confusing. This reviewer can not even complete an analysis on the example file provided in the web site, that is totally a negative case according to the aim of the tool for alleviating the burden for data analyais. Overall, there are several major concerns needed to be addressed. Comments are given in the follows.

  1. A detailed instruction on how to prepare the data file for analysis should be provided, especially how to identify the time/condition keywords in the website. They provide no clue on how they process the keywords making the analysis pipeline like a blakbox. Users are totally unaware how the input keywords affact their analysis. Some exmples should be given to help users to get familiar with the tool. The simple instruction on the website and a DEMO code is not helping.
  2. Follow the previous point, this reviewer tried several combinations of parameters in the website and most of the parameters output something, evan when an invalid time keyword was utilized. There is no idea what algorithm is behind the tool.
  3. Moreover, a tool to automatically varify the input file and generate possible keywords will be considred intuitive for use.
  4. An editing tool to facilitate the formating of data file or choosing the samples for comparison is strongly recommended. Please see GEO2R as an example. 
  5. Since the tool identifies also TF binding sites and miRNA targets, subsequent GO/pathway analysis on the two sets can provide more insights.
  6. There is no description on the inclusion/exclusion of the miRNA genes  in the analysis of GO/pathway.
  7. https protocol for ensuring the safety of connection should be implemented
  8. Please clarify the data processing and storage policy.

Author Response

(The authors gave the same response as above.)

Reviewer 3 Report

In this manuscript, authors developed a web application to analyse time course transcriptome data with multiple conditions. The manuscript is well written and the web application is easy to use. I have the following few minor comments

  1. Authors should provide an example link which should automatically load the sample data and show the sample results. This is a very basic addition in any web application nowadays.
  2. Instead of providing screenshots (or in-addition to this) of the web application, I strongly suggest to include a flowchart to clearly depicts the workflow of the web application from input to final output.

Author Response

(The authors gave the same response as above.)
